# Antimitochondrial Antibodies and Primary Biliary Cholangitis in Patients with Polymyalgia Rheumatica/Giant Cell Arteritis

**DOI:** 10.3390/medicina57040350

**Published:** 2021-04-06

**Authors:** Ciro Manzo, Maria Maslinska, Alberto Castagna, Elvis Hysa, Alfonso Merante, Marcin Milchert, Tiziana Gravina, Betul Sargin, Maria Natale, Carmen Ruberto, Giovanni Ruotolo

**Affiliations:** 1Azienda Sanitaria Locale Napoli 3 Sud, Internal and Geriatric Medicine Department, Rheumatologic Outpatient Clinic, Health District No. 59, Sant’Agnello, 80065 Naples, Italy; 2National Institute of Geriatrics, Rheumatology and Rehabilitation, 02-637 Warsaw, Poland; maslinskam@gmail.com; 3Azienda Sanitaria Provinciale Catanzaro, Primary Care Department, Casa Della Salute di Chiaravalle Centrale, Fragility Outpatient Clinic, 88100 Catanzaro, Italy; albertocastagna78@gmail.com; 4Research Laboratory and Academic Division of Clinical Rheumatology, Department of Internal Medicine, San Martino Policlinic Hospital, 16121 Genoa, Italy; elvis.hysa@gmail.com; 5Geriatric Unit, Azienda Ospedaliera “Pugliese-Ciaccio”, 88100 Catanzaro, Italy; merantealfonso@libero.it; 6Department of Rheumatology, Internal Medicine, Geriatrics and Clinical Immunology of Pomeranian Medical University, 71-871 Szczecin, Poland; marcmilc@hotmail.com; 7Hepatology Unit, Azienda Ospedaliera Universitaria “Mater Domini”, 88100 Catanzaro, Italy; tizianagravina@alice.it; 8Department of Physical Medicine and Rehabilitation, Division of Rheumatology, Medical Faculty of Adnan Menderes University, Aydin 09100, Turkey; betul.cakir@yahoo.com; 9Azienda Sanitaria Locale Napoli 3 Sud, Internal Medicine Department, Health District No. 58, Gragnano, 80054 Naples, Italy; maria.natale.int@alice.it; 10Azienda Sanitaria Provinciale Catanzaro, Primary Care Department, 88100 Catanzaro, Italy; albertocastagna@tiscali.it; 11Azienda Ospedaliera “Pugliese-Ciaccio”, Geriatric Medicine Department, 88100 Catanzaro, Italy; giovanni.ruotolo1@virgilio.it

**Keywords:** polymyalgia rheumatica, giant cell arteritis, antimitochondrial antibodies, primary biliary cholangitis, diagnostic and classification criteria, liver biopsy, cholestasis indices, narrative review

## Abstract

*Background and Objectives*: Laboratory liver abnormalities can be observed in patients affected with polymyalgia rheumatica (PMR) and/or giant cell arteritis (GCA), especially with a cholestatic pattern. The first objective of our review article is to discuss the potential link between antimitochondrial antibodies (AMA) and/or primary biliary cholangitis (PBC) and PMR/GCA, according to the evidences of literature. The second objective is to discuss the association of PMR/GCA with the other rheumatic diseases having PBC as a common manifestation. *Materials and Methods*: A literature search was performed on PubMed and Medline (OVID interface) using these terms: polymyalgia rheumatica, giant cell arteritis, antimitochondrial antibodies, primary biliary cholangitis, primary Sjogren’s syndrome, systemic sclerosis, and systemic lupus erythematosus. The search was restricted to all studies and case reports published in any language. Reviews, conference abstracts, comments, and non-original articles were excluded; however, each review’s reference list was scanned for additional publications meeting this study’s aim. When papers reported data partially presented in previous articles, we referred to the most recent published data. *Results and Conclusions*: Our literature search highlighted that cases reporting an association between AMA, PBC and PMR/GCA were very uncommon; AMA antigenic specificity had never been detected and biopsy-proven PBC was reported only in one patient with PMR/GCA. Finally, the association of PMR/GCA with autoimmune rheumatic diseases in which PBC is relatively common was anecdotal.

## 1. Introduction

Polymyalgia rheumatica (PMR) is considered the most common inflammatory rheumatic disease in the elderly. An age of over 50 is a diagnostic and classification criterion [1,2,3,4]. Typical in PMR patients is sudden-onset bilateral pain in the shoulder and pelvic girdles and morning stiffness lasting >45 min. Neck aching can be associated. Patients usually complain of significant restrictions in self-care activities of daily living (ADL). Additional symptoms, such as fever, general discomfort, fatigue, loss of appetite, and loss of weight, can be present in some patients [5,6]. PMR incidence increases until the age of 90, with a peak around the age of 75. As confirmed by a recent systematic literature review [7], PMR prevalence and incidence are influenced by study design and population. In particular, it is estimated an annual incidence rate between 0.12 and 2.3 cases/1000 persons aged 50 years and older [8,9,10,11,12]. 

In up to 20% of cases, PMR can be associated with giant cell arteritis (GCA), a chronic granulomatous vasculitis affecting the aorta and its branches. GCA patients, in turn, can have polymyalgic manifestations in up to 60% of cases [13]. GCA is the most common primary vasculitis in patients aged over 50 years [7,11]. Recently, a meta-analysis reported a pooled incidence of 10 cases and a pooled prevalence of 51.74 cases for 100,000 persons over 50 years old, highlighting that latitude correlated significantly with GCA incidence but not with prevalence [14]. Both PMR and GCA improve with glucocorticoid (GC) treatment [15].

Laboratory liver abnormalities are reported in PMR/GCA patients, especially with a cholestatic pattern, namely an increase of serum alkaline phosphatase (AP) due to the hepatic isoenzyme. Less frequent is a cytolytic pattern with elevation of alanine (ALT) and aspartate (AST) transaminases without raised AP and gamma-glutamyltranspeptidase (gGT) [16]. Some investigators hypothesized a possible immunological pathogenesis and, more specifically, reported an association with primary biliary cholangitis (PBC) [17], an autoimmune liver disease characterized by the damage of intrahepatic bile ducts. More than 95% of patients with PBC are positive for antimitochondrial antibodies (AMA). Incidence of PBC ranges from 0.33 to 5.8 per 100,000 persons, with the reported point prevalence ranging from 1.91 to 33.8 per 100,000 persons. As for PMR/GCA, Europe and North America have the highest reported prevalence of PBC worldwide [18]. AMA can be identified in patients affected with PMR/GCA, in absence of diagnosed PBC [19].

Finally, PMR/GCA can be associated with some autoimmune rheumatic diseases, such as primary Sjogren’s syndrome (pSS), systemic sclerosis (SSc) and systemic lupus erythematosus (SLE), in which PBC is more commonly recognized and expected. Indeed, patients affected with PBC have at least a 60% probability of having an autoimmune extra hepatic condition, with rheumatologic among these [20].

**Objectives**: the first aim of our review article was to discuss the relationship between PMR/GCA and AMA/PBC, according to the evidence in the literature. A secondary aim was to examine if the association of PMR/GCA with pSS/SLE/SSc is a risk factor for PBC and/or AMA.

## 2. Materials and Methods

A narrative non-systematic review (PRISMA protocol not followed) was performed on PubMed and Medline (OVID interface) with these terms: polymyalgia rheumatica, giant cell arteritis, antimitochondrial antibodies, primary biliary cholangitis, primary Sjogren’s syndrome, systemic sclerosis, and systemic lupus erythematosus. The search was restricted to all studies and case reports published in any language. Reviews, conference abstracts, comments, and non-original articles were excluded; however, each review’s reference list was scanned for additional publications meeting this study’s aim. When papers reported data partially presented in previous articles, we referred to the most recent published data.

## 3. Results

According to our literature search, the cases reporting an association between AMA, PBC and PMR/GCA are very uncommon.

In 1970, Walker et al. first reported the presence of AMA in serum of a female patient who suffered with generalized muscle pain responsive to GC therapy. PMR was diagnosed, but this diagnosis is at least questionable, not having been supported by any specific diagnostic criteria. The patient’s liver biopsy showed abnormal findings, compatible with generic autoimmune hepatitis. Walker et al. stated that, since there was no evidence of bile duct necrosis from the liver biopsy, a diagnosis of PBC could not be supported, while emphasizing that patient had clinical and biochemical features suggestive of PBC [21].

In 1978, Robertson et al. reported three female patients with PMR and suggestive symptoms and laboratory abnormalities evocative of PBC. In particular, two out of these three patients showed high titers of AMA; in one patient, elevated serum concentrations of anti-smooth muscle antibodies (ASMA) were also present. A needle biopsy was performed only on the patient who was AMA negative and showed a multilobular cirrhosis with heavy infiltration of the portal tracts by chronic inflammatory cells. Bile thrombi were not found [18]. In the same year, other investigators reported that they found no patients with PMR among 83 suffering from biopsy-proven PBC [22].

In Sattar et al.’s case series, published in 1984 [19], AMA were detected in the sera of 11 (four with low titers) out of 36 patients with a diagnosis of PMR or GCA. To date, this is the broadest case series in published literature regarding the association between AMA, PBC and PMR/GCA. Of the 10 patients affected with PMR in which AMA were detected, GCA was associated in none. The authors detected no correlation between AMA titers, inflammatory indexes and clinical state. Liver biopsy was performed only in a patient having an AMA title of 1:2500 and revealed nonspecific histological findings consistent with a mild hepatitis. The authors hypothesized that AMA positivity in PMR patients could be the reflection of a subclinical PBC or alternatively that it could be reactive to an extra-hepatic mitochondrial damage. With regard to the second hypothesis, the possibility that PMR can coexist with an acquired mitochondrial myopathy has been reported in the published literature [23]. In addition, mitochondrial damages have been described in electron microscopically examined skeletal muscles from patients with clinical confirmed PMR [24]. In Sattar et al.’s case series, a patient also had GCA associated with SS, a connective tissue disease that links to PBC as part of a systemic autoimmune epithelitis [25]. In this patient, AMA were present at low titers (1:20) [19].

In 1988, Gagnerie et al. reported a patient affected with PBC associated with GCA and PMR [26].

More recently, Eginli et al. reported a 59-year-old Caucasian woman affected with frontal fibrosing alopecia associated with PBC and PMR [27]. However, in their case report, it was unclear how PMR and PBC were diagnosed.

The aforementioned case series are schematically listed in Table 1.

An association of PMR/GCA with pSS, SSc and SLE has been very scarcely reported in published literature. The most common association we found is with the pSS. A retrospective study compared the clinical and laboratory findings of 16 patients affected with pSS associated with PMR and a large cohort of 531 patients affected by pSS without PMR. In this study, the prevalence of PMR among pSS patients was four time more frequent than in the general population. PMR developed several years after the diagnosis of pSS. While the entire group with pSS in cited studies was relatively large, the subgroup with PMR was relatively small; this excluded its broader analysis and the extrapolation of its characteristics, including the presence of AMA and PBC [28]. To date, only the patient reported by Sattar et al. [19] had low titers AMA associated with GCA and pSS.

Late-onset SLE presenting as PMR has been reported in nine patients; in none of these were AMA or PBC identified [29,30,31,32]. Association with SSc has been reported only in very few patients affected with GCA (without PMR) [33,34,35,36].

## 4. Discussion

PBC, PMR and GCA have different immune-mediate pathogenetic mechanisms. For instance, while in PBC plasma cells produce antibodies as a response to an environmental mimic of the mitochondrial subunit E2, in PMR and GCA there is no auto-antibodies production. Moreover, the cells present in the inflammatory infiltrates in the affected synovial membranes of joints, bursae and inflamed arteries of patients with PMR/GCA are different from the cells present in the inflammatory infiltrates in portal areas and bile ducts in patients with PBC. Indeed, while in PMR and GCA they are macrophages, adventitial dendritic cells and T lymphocytes (especially T_H_17), in PBC the protagonists are IgM+ plasma cells, CD8+ lymphocytes, natural killer cells, mucosal-associated invariant T cells, and fibroblasts. Likewise, granulomas observable in PBC and in GCA display different features. In particular, PBC granulomas are mainly composed of epithelioid cells displaying features of immature dendritic cells, whose maturation is potentially inhibited by the IgM synthetized from the surrounding plasma cells. In GCA granulomas, instead, CD4 T cells and highly activated macrophages are characteristically present. These cells often fuse together, forming multinucleated giant cells [37,38]. According to our literature search, in patients suffering from PMR/GCA and PBC in which liver biopsy was performed, no common immunological link was found.

Liver involvement with raised cholestasis biomarkers can be detected in PMR/GCA patients. This does not mean that PBC is present. In line with the European Association for the study of the liver (EASL), diagnosis of PBC can be made based on elevated AP and the presence of AMA at titer >1:40 [39]. Indeed, as reported by Zein et al. in AMA-positive patients with AP elevation >1.5 times and aspartate transaminases elevation >5 times the upper laboratory values, the positive predictive value was proven to be 98.2%, reaching 100% once pertinent clinical variables (such as itching) were included in the analysis [40]. In the context of AMA-negative patients, specific ANA should be tested: sp100 and gp210. EASL recommend against liver biopsy for the diagnosis of PBC unless PBC-specific antibodies are absent or coexistent autoimmune hepatitis (AIH), non-alcoholic steatohepatitis (NASH) or other co-morbidities are present [39]. The possibility that GCs can realize a confounding factor should be considered. Indeed, abnormal liver function tests typically normalize after GC therapy [17], and this can delay diagnosis of PBC. To the best of our knowledge, biopsy-proven PBC has been reported only in one patient with PMR/GCA. In this patient, PBC was diagnosed many years before the onset of Horton’s disease [19].

Manifestations such as pruritus (absent in patients with PMR/GCA and present in roughly 50% of patients with PBC) and/or fatigue might be additional warning signals for suggesting the research of AMA.

As previously mentioned, more than 95% of patients with PBC are positive for AMA [41,42]. However, AMA can also be present in healthy adults [43]. Its prevalence in the general population has been reported to range from 0.07% to 9.9% [44]. On the other hand, we must also consider that AMA positivity could be reactive to an immunological stimulation driven from the rheumatic disease and not be an expression of PBC [45,46]. Some investigators analyzed retrospectively 100 patients who were AMA test positive: 61 suffered from liver diseases (36 PBC) and 39 from non-liver diseases. Among these 39 patients, nine had systemic autoimmune diseases, and three were affected with organ-specific autoimmune diseases. Very high (>10,000 U/mL) and high (1001–10,000 U/mL) anti-M2 values were present mainly in patients with liver disease; medium (101–1000 U/mL) and low (5–100 U/mL) values were present in patients with extra-hepatic disease [47]. In line with this study, AMA titers are closely linked to liver or to extra-hepatic disease. According to our literature search, only four patients had high AMA values (1280–8000 U/mL), and one of these had also ASMA high titer.

Finally, in some patients affected with PBC, identification of AMA is not possible. The ratio of AMA-negative/AMA-positive PBC is estimated to be 1/9. However, in nearly all AMA-negative PBC patients, it is possible to find other PBC-specific antinuclear antibodies, such as anti-sp100 and anti-glycoprotein-210 antibodies, confirming PBC as an autoimmune disease [48]. In 2008, a comparative study highlighted that AMA-negative PBC patients are similar to AMA-positive patients in clinical and laboratory findings, percentage of regulatory T cells (Tregs) in peripheral blood, liver biopsy features and response to treatment. Based on these findings, AMA-negative PBC was proposed as a variant of AMA-positive PBC rather than as a separate clinical entity [49]. According to our literature review, data regarding AMA-negative PBC in PMR/GCA patients are anecdotal.

Finally, the association of PMR/GCA with the autoimmune rheumatic diseases in which PBC is relatively common is very scarcely reported in published literature. In particular, some researchers proposed to add pSS to the list of PMR-mimicking syndromes [50]. Chronic fatigue, arthralgia, myalgia and even a feeling of dryness are common in both pSS and PMR. This can be confusing, and patients with these symptoms may be misdiagnosed [51].

Moreover, a large number of pSS patients are over 50 years old (so-called “Elderly-onset primary Sjogren’s syndrome—EoPSS”) [52,53] and can have overlapping of PMR symptoms. Table 2 shows similarities and differences between PMR and EopSS.

## 5. Conclusions

AMA and PBC have been not commonly reported in patients affected with PMR/GCA. Moreover, as our literature search highlighted, the available data were often incomplete. 

To date, the possibility that it can be an association rather than a mere coincidence is still speculative. Therefore, further studies (especially histopathological studies) should be planned in order to clarify if PMR and GCA can be the “authors” of AMA positivity and/or PBC.

Even if liver biopsy is not always mandatory for PBC diagnosis, a deepening with needle biopsies in patients with PMR/GCA when AP and AMA are raised could be useful in order to identify, on a large scale of case studies, a potential immunological involvement of the liver.

## Figures and Tables

**Table 1 medicina-57-00350-t001:** Case series reporting an association between PMR/GCA and AMA positivity/PBC.

Number of Cases	Diagnosis Made	Clinical Signs and Symptoms/Diagnostic Criteria	Liver Laboratory Abnormalities	AMA Positivity	Liver Biopsy
1 [21]	PMR	GC-responsive generalized muscle pain	Increase of AP and γGT	Yes	Infiltration of lymphocytes, macrophages and occasional plasma cells
1 [18]	PMR	Inflammatory pain in the neck, shoulders and low back with increased inflammatory indexes and responsiveness to GC treatment	Increase of AP and γGT	Yes	Not performed
1 [18]	PMR	Shoulder pain, weight loss and increased inflammatory indexes	Increase of AP and AST	No	Not performed
1 [18]	PMR	Hip and shoulder pain, MS and increased inflammatory markers	Increase of AP, γGT and AST	Yes	Not performed
9 [19]	PMR	Diagnostic criteria: limb girdle pain, MS, and muscle tenderness in the absence of polyarthritis or myopathy	Increase of ALP	Yes	Not performed
1 [19]	PMR	Diagnostic criteria: limb girdle pain, MS, and muscle tenderness in the absence of polyarthritis or myopathy	Increase of ALP	Yes	Mild inflammatory aspecific changes
1 [19]	GCA + pSS	TA biopsy-proven	Increase of AP	Yes	Not performed
1 [26]	PMR+GCA	Girdle pain, recent-onset headache, weight loss and increased inflammatory indexes; TA biopsy was performed	Increase of AP and γGT	Yes	Portal fibrosis, mononuclear cells infiltrate around the bile ducts
1 [27]	PMR+PBC	Previous diagnosis of PMR and PBC (unclear criteria) in patient affected by fibrosing alopecia	Unknown	Unknown	Unknown

PMR = Polymyalgia rheumatica; GCA = giant cell arteritis; pSS = primary Sjogren’s syndrome; PBC = primary biliary cholangitis; TA = temporal artery; GC = glucocorticoid; MS = morning stiffness; AP = alkaline phosphatase; γGT = gamma-glutamyl transpeptidase; AST = aspartate aminotransferase.

**Table 2 medicina-57-00350-t002:** PMR and EopSS: similarities and differences.

PMR	EopSS
SIMILARITIES	
females > males	females > males
mean age > 50 y.o.	mean age > 50 y.o.
fatigue	fatigue
arthralgia	arthralgia
peripheral arthritis usually non-destructive	arthritis usually non-destructive
muscle weakness	muscle weakness
elevated ESR	elevated ESR
often decreased vitamin D level	often decreased vitamin D level
myopathy in a course of PMR, GCS treatment and/or hypothyroiditis (Hashimoto’s disease)	myopathy in a course of disease, GCS treatment and/or hypothyroiditis (Hashimoto’s disease)
risk of transition in RA	overlap syndrome with RA or RA and secondary SS
dryness (eyes, mouth) in elder group of patients (age-related)	dryness (eyes, mouth)—typical symptom
DIFFERENCES	
elevated CRP	usually CRP in normal range
bursitis	not typical
arthralgia+arthritis	arthralgia > arthritis
large vessel vasculitis	medium and small vessel vasculitis
ANA may be positive in low titer	ANA positive, anti-rybonucleitide antibodies (anti-Ro 60, anti-Ro52, anti-La)
probability of co-existence of Hashimoto’s disease	often co-existence of autoimmune thryroiditis (Hashimoto’s disease)
screening for neoplasms is obligatory, especially in older group or in patients with risk factors	elevated risk of lymphoma development

ANA = antinuclear antibodies; CRP = C reactive protein; ESR = erythrocyte sedimentation rate; RA = rheumatoid arthritis; EopSS = elderly onset primary Sjogren’s syndrome.

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
