# Peer review of "Antimitochondrial Antibodies and Primary Biliary Cholangitis in Patients with Polymyalgia Rheumatica/Giant Cell Arteritis"

_medicina, 2021, doi:10.3390/medicina57040350_

Round 1
Reviewer 1 Report
In This review, Manjo et al reviewed the published literature to find an association between anti-mitochondrial antibodies primary cholangitis and polymyalgia rheumatica. By scrutinizing the case report and studies the authors could not find a clear association between these characters although some studies reported the finding of AMA or PBC in PMR patients. The authors aimed to highlight a lacuna in literature which might be overlooked by clinicians/researchers when studying the clinical parameters and draw attention towards these two characters in PMR. However, I have the following comments.
1-In the present form of the review its not clear why the authors chose to analyze AMA and PBS in PMR. The authors should explain in detail why AMA and PBC are important parameters of PMR. They should also discuss now analyzing AMA and PBS in PMR would be helpful for further diagnosis or developing therapeutics.
2- In the line 102 the word titles need to be replaced by titers.
Author Response
Dear Reviewer,
we thank You so much for Your comments and suggestions.
In the attached file, You can read our point-by-point answers.

Reviewer 2 Report
The manuscript has a poor assessment and should be rewritten.
- Title: please change: two characters in search of an author is wrong
- The author should keep into account the the following issues: i) epidemiology of polymyalgia; ii) epidemiology of GCA; iii) epidemiology of both; iv) epidemiology of PBC; v) prevalence of AMA and PBC in polymialgia and eventually in polymialgia/GCA; vi) prevalence of polymyalgia and GCA in PBC
- Figure: is mostly incorrect: abdominal pain is not a peculiar symptom of PBC; itching is present in roughly 50% of patients with PBC; besides itching, fatigue is a peculiar symptom of PBC; In case of diagnostic suspect for PBC in AMA-ve subjects sp100 and gp210 should be performed; What do you mean with aspecific imaging?; see the comment regarding liver biopsy.
- Page 6, line 166 "Therefore, when itching and abdominal pain..." EASL guidelines recommend that diagnosis of PBC can be made based on elevated ALP and the presence of AMA at titre >1:40; in the context of AMA-ve patients specific ANA should be tested: sp100 and gp210. EASL recommend against liver biopsy for the diagnosis of PBC unless PBC-specific antibodies are absent, coexistent AIH or NASH or other co-morbidities are present.
- Conclusions (second paragraph): the first sentence is a nonsense: a mere coincidence is incorrect. You can speculate on an association between two or more autoimmune conditions that share the same pathogenic mechanism.
- Reference are incorrect: several references are lacking (i.e. Floreani Best Pracr Res Clin Gastroenterol 2018) and often some references are not reported in the text (i.e. reference n.25)
Author Response
Dear Reviewer,
many thanks for Your valuable comments and suggestions.
Please, read the attached file.
Best regards.

Round 2
Reviewer 2 Report
The present revision is suitable for publication